# Development and Application of an Automated Raman Sensor for Bioprocess Monitoring: From the Laboratory to an Algae Production Platform

**DOI:** 10.3390/s23249746

**Published:** 2023-12-11

**Authors:** Wiviane Wieser, Antony Ali Assaf, Benjamin Le Gouic, Emmanuel Dechandol, Laura Herve, Thomas Louineau, Omar Hussein Dib, Olivier Gonçalves, Mariana Titica, Aurélie Couzinet-Mossion, Gaetane Wielgosz-Collin, Marine Bittel, Gerald Thouand

**Affiliations:** 1Nantes Université, CNRS, Oniris, GEPEA, UMR CNRS 6144, F-85000 La Roche-sur-Yon, France; wiviane.wieser@univ-nantes.fr (W.W.); thomas.louineau@univ-nantes.fr (T.L.); omar.dib@univ-nantes.fr (O.H.D.); gerald.thouand@univ-nantes.fr (G.T.); 2Tronico-Alcen, 26 rue du Bocage, F-85660 Saint-Philbert-De-Bouaine, France; mbittel@tronico-alcen.com; 3Nantes Université, Plateforme Algosolis, UMS CNRS 3722, F-44600 St Nazaire, France; benjamin.legouic@univ-nantes.fr (B.L.G.); emmanuel.dechandol@univ-nantes.fr (E.D.); laura.herve@univ-nantes.fr (L.H.); 4Nantes Université, CNRS, Oniris, GEPEA, UMR CNRS 6144, F-44600 St Nazaire, France; olivier.goncalves@univ-nantes.fr (O.G.); mariana.titica@univ-nantes.fr (M.T.); 5Nantes Université, ISOMer, UR 2160, F-44000 Nantes, France; aurelie.couzinet-mossion@univ-nantes.fr (A.C.-M.); gaetane.wielgosz-collin@univ-nantes.fr (G.W.-C.)

**Keywords:** optical sensor, Raman spectroscopy, microalgae, pilot scale, monitoring

## Abstract

Microalgae provide valuable bio-components with economic and environmental benefits. The monitoring of microalgal production is mostly performed using different sensors and analytical methods that, although very powerful, are limited to qualified users. This study proposes an automated Raman spectroscopy-based sensor for the online monitoring of microalgal production. For this purpose, an in situ system with a sampling station was made of a light-tight optical chamber connected to a Raman probe. Microalgal cultures were routed to this chamber by pipes connected to pumps and valves controlled and programmed by a computer. The developed approach was evaluated on *Parachlorella kessleri* under different culture conditions at a laboratory and an industrial algal platform. As a result, more than 4000 Raman spectra were generated and analysed by statistical methods. These spectra reflected the physiological state of the cells and demonstrate the ability of the developed sensor to monitor the physiology of microalgal cells and their intracellular molecules of interest in a complex production environment.

## 1. Introduction

Microalgae are an extremely diverse group of photosynthesising microorganisms. They are rich in pigments, proteins and other valuable metabolites [1]. The growth of microalgae relies on accessible inorganic resources that enable them to live in a wide range of habitats [2]. They therefore represent a promising feedstock, as they do not necessarily compete with food crops [3]. Due to their economic interest, microalgae have attracted a great deal of attention from industries demanding products based on sustainable resources [4,5,6]. However, uncontrolled changes in microalgal culture parameters can trigger changes to cell metabolism and alter the productivity of this process, potentially making it unprofitable. This type of risk highlights the importance of monitoring microalgal bioprocesses [7].

The methods most commonly used to monitor microalgal production are mono-parametric. These allow the real-time monitoring of specific parameters that can affect algal growth and productivity (e.g., pH, temperature, pCO_2_, pO_2_, biomass, etc.). However, because they only measure a single parameter, multiple sensors are required to collect complete information on a system. This can be cumbersome and may not provide a comprehensive system overview [8,9]. Alternatively, multiparametric sensors can be used to measure multiple parameters simultaneously, which can be more efficient and provide a more comprehensive view of a system. These sensors can make bioprocesses more profitable by minimising product losses and improving cell viability [10,11]. Among the available methods, Raman spectroscopy is a promising technology offering rapid, non-invasive and non-destructive measurement of living cells [12]. This technique, based on inelastic light scattering, measures the vibrational modes of chemical bands, whose positions and intensities provide information about biomolecules [13,14]. The obtained spectra reflect the molecular components of microalgal cells like lipids, proteins, carotenoids and chlorophylls. Raman spectroscopy offers advantages for analysing microalgal cells directly in their culture environment without any interference from the water. The wide range of excitation wavelengths available makes it possible to avoid damaging microalgal cells or perturbing the Raman signature by the fluorescence emissions of some molecules [15,16,17].

A large number of studies have shown the ability of confocal Raman spectroscopy to measure molecules of interest produced in microalgae, such as carbohydrates, pigments and lipids [18,19,20,21]. In some previous studies, authors have made original applications of Raman spectroscopy to identify microalgal species [22,23,24] or as a multiparametric tool to monitor cell physiology during their production [25,26,27,28]. However, most of these studies have mainly focused on off-line Raman analysis at a laboratory-scale, confirmed by conventional analytical techniques such as mass spectrometry, nuclear magnetic resonance or chromatographic methods [9,27,29,30]. The possibility of increasing the accuracy of Raman spectroscopy to identify some algae species at the single-cell level was reported [25,27,31,32]. Nevertheless, the most of these studies are limited to laboratory use due to the necessity to deposit algae on a suitable surface. On the other hand, the literature reveals the online monitoring of microalgae by Raman probes immersed directly into the liquid cultures [33,34,35,36]. These very interesting investigations demonstrate the ability of Raman spectroscopy to monitor some specific molecules in situ. However, all these studies focused on a bench-scale proof of concept without testing the methods on a large-scale over a long period of time. Some issues such as biofilm formation on the probe, the biological contamination or light interference were also not properly addressed [34,37,38,39]. Technical details concerning the washing cycles or the automation of analysis have so far been neglected. The challenges that remain are therefore to automate such processes, adapt the technology to be economically viable under the field conditions of the production of microalgae and to scale it up to an industrial level. To achieve these objectives, the entire optical approach, including optical reading, statistical analysis and sampling, needs to be optimised for the widespread large-scale implementation of spectroscopy. One crucial aspect is to ensure that the materials used comply with industry standards and quality. The system should also preferably be shielded from light to enhance the adaptation of spectral analysis to the lighting conditions encountered in the microalgae industry. Additionally, it should be autonomous to ensure the reproducibility of the analyses and to minimize human intervention [40]. Considering the limitations of previous studies and the remaining challenges, this study proposes an automated Raman approach and its integration into a semi-industrial microalgal platform. By focusing on the specific requirements and complexities of large-scale implementation, this study aims to bridge the gap between bench-scale proof-of-concept and industrial problem solving, thereby contributing to the advancement of microalgae-based industries and enabling the more efficient monitoring of bioprocesses.

## 2. Materials and Methods

### 2.1. Microalgal Strain and Culture Conditions

*Parachlorella kessleri* strain 2229 was bought from the University of Austin, Texas (UTEX) culture collection. The microalgae were cultivated in sterile modified Bold Basal Medium (BBM) with the following composition (in mM): MgSO_4_⋅7H_2_O 9.1 × 10^−1^ (Fisher Bio, Illkirch, France, ref. 10553335), CaCl_2_⋅2H_2_O 1.7 × 10^−1^ (Acros Organics, Coueron, France, ref. 10158280), ZnSO_4_⋅7H_2_O 7.72 × 10^−4^ (Sigma-Aldrich, Darmstadt, Germany, ref. 221376), Co(NO_3_)_2_⋅6H_2_O 1.51 × 10^−4^ (Sigma-Aldrich, Darmstadt, Germany, ref. 131258,1209), CuSO_4_ 4.95 × 10^−4^ (Fisher Scientific, Illkirch, France, ref. A4778701), H_3_BO_3_ 4.63 × 10^−2^ (Fisher Scientific, Illkirch, France, ref. A4703751), MnCl_2_⋅4H_2_O 9.15 × 10^−3^ (Sigma-Aldrich, Darmstadt, Germany, ref. M8054), Na_2_MoO_4_ 1.06 × 10^−3^ (Alfa Aesar, Massachusetts, Haverhill, MA, USA, ref. 01221422.22), C_10_H_14_N_2_O_8_Na_2_⋅2H_2_O 1.34 × 10^−1^ (Sigma-Aldrich, Darmstadt, Germany, ref. E5513), FeSO_4_⋅7H_2_O 5 × 10^−2^ (Labosi, Illkirch, France, ref. A4809001), K_2_HPO_4_ 8.61 × 10^−1^ (Sigma-Aldrich, Darmstadt, Germany, ref. 795496), KH_2_PO_4_ 9 × 10^−1^ (Acros Organics, Coueron, France, ref. 42420500), NaHCO_3_ 1.5 × 10^−2^ (Merck, Darmstadt, Germany. 106329) and NaNO_3_ 8.82 (Sigma-Aldrich, Darmstadt, Germany, ref. S5506).

The inoculums were grown in Erlenmeyer flasks (500 mL) in an incubator (Innova 42, Eppendorf, New Brunswick Scientific, Enfield, CT, USA) and monitored for 21 days at 23 °C and 130 rpm under continuous light (incident photon flux density of 30 µmol·m^−2^·s^−1^).

At the laboratory scale, the photobioreactor (PBR) used was a 1-L airlift type with a culture depth Lz = 30 mm (perpendicular to the optical surface). The PBR was constructed of transparent polymethyl methacrylate (PMMA) with a stainless-steel (type 316L) support coupled to a polypropylene plate coil panel. The temperature was maintained by a flow of water from a cryothermostat (Lauda RC20, Königshofen, Germany) at 23 °C (±0.1°) running across the plate coil panel. Continuous illumination was provided by a 40 × 60 cm white LED light panel (LED Sunlight-Z1, Bionef, Montreuil, France) with a photon flux density of 120 µmol·m^−2^·s^−1^. The pH was regulated at 7.5 ± 0.15 by a CO_2_ injection-based autoregulation system performed by solenoid valves (SMC VX220HL, Yorba Linda, CA, USA) and controlled by a transmitter (Mettler Toledo M300, Columbus, OH, USA). The air bubble inlet was created by an air pump (Air 2x180, Inwa, Joinville-Le-Pont, France). The air flow ensures the homogenisation of the culture and prevents the algal cells from settling at the bottom of the PBR. Cells were cultivated in a nitrogen-modified BBM with a concentration of 8.82 mM NaNO_3_ during their growth phase. On the tenth day, the culture underwent centrifugation (Avanti series J-15, Beckman Coulter, Pasadena, CA, USA) at 4400× *g* for 20 min and were then washed with nitrogen-free BBM. This washing procedure was carried out twice. Subsequently, the cells were resuspended in the nitrogen-free BBM and transferred to the photobioreactor under sterile conditions. Notably, cultivation in nitrogen-free BBM stimulates intracellular lipid production.

At the pilot scale, the experiments were conducted at the AlgoSolis R&D facility in a 100-L Air Lift Tubular photobioreactor (Synoxis algae, Le Cellier, France) at pH 7.5 ± 0.15 in a greenhouse. pH was regulated by a system based on CO_2_ injection using a Mettler Toledo transmitter M200. During the first six days, the microalgal cells were grown in 50 L of BBM containing an initial NaNO_3_ concentration 50% lower than the standard concentration (4.41 mM). Another 50 L of NaNO_3_-free BBM was then added. To limit the problem of heterogeneity, the photobioreactor was under continuous stirring by agitator impellers and controlled air bubbling. The algal culture was derived from the bottom of the photobioreactor and sent to the measuring optical chamber.

### 2.2. Determination of the Cell Density and the Concentration of Total Chlorophyll and Carotenoids

Growth was monitored daily by measuring the absorbance at 650 nm using a UV spectrophotometer (SAFAS UVmc2, Monte Carlo, Monaco) [41,42]. For this, a volume of 1 mL of microalgae was introduced into a disposable polystyrene cuvette (Brand GmbH, Wertheim, Germany). The cultures were diluted until the optical density was less than 0.8. The actual optical density was then determined by multiplying by the dilution factor.

Concentrations of chlorophyll a and b and total carotenoids were measured daily by a spectrophotometric method [43]. A volume of 0.5–1.5 mL of microalgae was centrifuged for 10 min at 13,400 rpm (MiniSpin plus centrifuge, Eppendorf, Hamburg, Germany). The supernatant was discarded, and the cells were re-suspended in 1.5 mL methanol 99.9% (Fisher Scientific, Illkirch, France, ref. M405615). Samples were incubated in the dark at 44 °C for 45–180 min to allow complete extraction, followed by 10 min of centrifugation at 13,400 rpm. The absorbance of these samples was measured on a spectrophotometer (SAFAS UVmc2, Monte Carlo, Monaco) at 480 nm (A_480_), 652 nm (A_652_) and 665 nm (A_665_) and was corrected by subtracting the turbidity (750 nm absorbance). The concentrations of chlorophyll a and b were calculated according to Equations (1) and (2). The total carotenoid concentration was calculated according to Equation (3) [43,44]:

Concentration of chlorophyll a in µg·mL^−1^
(1)$$C=\frac{1.5}{V_{biomass}}\left(-8.0962A_{652}+16.5169A_{665}\right)$$

Concentration of chlorophyll b in µg·mL^−1^
(2)$$C=\frac{1.5}{V_{biomass}}\left(27.4405A_{652}-12.1688A_{665}\right)$$

Concentration of carotenoids in µg·mL^−1^
(3)$$C=\frac{1.5}{V_{Biomass}}\left(4A_{480}\right)$$

### 2.3. Lipid Extraction and Analysis

Lipid extraction was performed in triplicate on a filtered biomass sample of 300 mL, using a modified method [45]. The samples were macerated in Erlenmeyer flasks on an orbital shaker (Edmund Bühler GmbH, SM-30) at room temperature. For the samples taken on the first six days, the biomass was macerated in a solvent mixture of dichloromethane (Sigma-Aldrich, Darmstadt, Germany, ref. 650463) and methanol (Sigma-Aldrich, Darmstadt, Germany, ref. 439193) (1:1 *v*/*v*) in a volume of 63 mL per gram of biomass for 24 h. The solvent volume was increased to 250 mL for the subsequent samples. Organic phases were then washed by addition of 20 mL of water, combined and dried over anhydrous Na_2_SO_4_ (Sigma-Aldrich, Darmstadt, Germany, ref. 239313) before evaporation on a rotary evaporator (Laborota 4000 efficient, Heidolph, Germany) to obtain the mass of lipid extract. Total lipid concentration was calculated with Equation (4).

Concentration of total lipid in mg·L^−1^
(4)$$C=\frac{masse\;lipid\;extract\;\left(mg\right)}{Sample\;Volume\;\left(L\right)}$$

Lipids were separated into classes by open silica gel column chromatography using 300 mg silica and CH_2_Cl_2_ in a Pasteur pipette. Neutral lipids (NL) were eluted with CH_2_Cl_2_, glycolipids (GL) with acetone (Sigma-Aldrich, Darmstadt, Germany, ref. 270725) and phospholipids (PL) with CH_3_OH (Sigma-Aldrich, Darmstadt, Germany, ref. 90964). The solvents were evaporated under nitrogen flux at 50 °C. The samples were then weighed to determine the amounts of NL, PL and GL.

The fatty acids were converted to fatty acid methyl esters (FAMEs) by transmethylation of 1–2 mg of crude extract (4 h at 80 °C, 500 µL CH_3_OH/HCl, 100 µL CH_3_OH, 100 µL CHCl_3_). The FAMEs were then converted to *N*-acyl pyrrolidines (NAP) using pyrrolidine (Sigma-Aldrich, Darmstadt, Germany, ref. P73803)/acetic acid (Sigma-Aldrich, Darmstadt, Germany, ref. A6283) (5:1, *v*/*v*). The separations of FAMEs and NAPs were achieved using a GC-MS instrument (Hewlett Packard HP 7890—GC System, Wilmington, DE, USA) equipped with a SLB-5^TM^ column (60 m × 0.25 mm × 0.25 µm) and linked to a mass detector (HP 5975C—E.I. 70 eV). Analyses were conducted under a constant flow rate (Helium-1 mL·min^−1^). The injector and detector temperatures were set at 250 and 280 °C, respectively, and 1 µL was injected in splitless mode. For FAME analyses, the initial temperature of the GC oven was held at 170 °C for 4 min, with a subsequent increase (3 °C·min^−1^) to 300 °C. For NAP analyses, the oven temperature was held at 200 °C for 4 min, then increased to 310 °C (3 °C·min^−1^) and maintained at this temperature for 20 min.

### 2.4. Determination of Nitrate Concentration

To monitor the levels of sodium nitrate in the BBM during the microalgae culture, the nitrogen-nitrate concentration in the 1-L lab-scale culture was determined using a method with NitraVer X Nitrogen-Nitrate Reagent (Method 100 Hach Lange, Loveland, CO, USA) based on the chromotropic acid method. The samples with reagents were measured by portable colorimeter (DR 900, Hach Lange, Loveland, CO, USA) with a reading range of 0–30 mg·L^−1^ NO_3_-N. Because of this range, samples containing nitrate levels exceeding 30 mg·L^−1^ were diluted by a factor of 5 or 10. At pilot scale, NO_3_^−^ monitoring was carried out using Quantofix test strips (Macherey-Nagel, Düren, Germany, ref. 91313) with a reading range of 10–500 mg·L^−1^ NO_3_^−^.

### 2.5. Development of the Raman Spectroscopy Measurement Approach

#### 2.5.1. Experimental Set-Up

An in situ sampling loop approach was developed for a Raman measuring device to adapt it to the industrial environment (Figure 1). For this purpose, a Raman probe was inserted into an optical flow-through cuvette (the measurement chamber). The chamber was constructed of 316L stainless-steel with dimensions of 70 mm × 40 mm × 30 mm. The roughness was 0.4 μm Ra and the inner geometry elliptical. This material was selected for its low surface microalgae adhesion, as described by Gross et al. [46] and following food industry standards [47].

The prototype was constructed using a container measuring 359 mm × 600 mm × 450 mm (Coffrets 19” Linkeo fixes, Legrand, France) capable of accommodating two in situ sampling loop approaches for parallel use. The interior of the container housed five solenoid valves, silicon pipes (∅3 mm inner × ∅5 mm outer) and a peristaltic pump (9QQ Peristaltic pump, Boxerpumps, Germany) for each system. These components facilitated the automated circulation of microalgae, as well as that of sterilized distilled water. Circulating water post-analysis enabled system flushing, eliminating potential cellular agglomerations within the prototype. The flow rate of the microalgae was set at 10 mL·min^−1^ for 1 h and 30 min, while the flow rate of the washing water was adjusted to 22 mL·min^−1^ for 30 min. To prevent the drying of any residual cellular debris on the optical probe or the chamber walls, the measurement chamber was kept filled with water between analyses. To control the peristaltic pump and solenoid valves, the prototype was equipped with a control station comprising an Arduino Mega 2560 microcontroller programmed using the Arduino Integrated Development Environment (IDE). The structural supports of the system were produced in polylactic acid using a 3-D printer (Ultimaker 2 + Connect, Ultimaker, Zaltbommel, The Netherlands).

#### 2.5.2. Raman Spectrometer and Parameters of Acquisition

The online monitoring experiments were carried out using an RA100 Renishaw Raman spectrometer with an integrated 532 nm laser and two compact fibre optic probes (RP10 Renishaw, New Mills, United Kingdom) driven by the WireTM 4.3 software package (Renishaw, UK). These probes were designed with a 3/8-inch diameter and a focal length of 3 mm. Additionally, they feature sapphire optical windows at their ends and were constructed with stainless steel bodies. The spectral resolution of this Raman spectrometer is 10 cm^−1^ in the range of 300–3100 cm^−1^ with a grating of 1800 lines/mm. The Raman laser has a nominal output power of 50 mW. The exposure time was 1 s or 10 s, varying depending on the physiological state of the microalgae according to Lieutaud et al. [48].

At the laboratory scale, 50 spectra were successively acquired daily for 36 days, obtaining a database of 1793 spectra of *P. kessleri*, using *WireTM 4.3*. At the pilot scale, the automation described above allowed the analyses to be carried out four times a day, separated by six hours, for 14 days, resulting in a database containing 2720 spectra of *P. kessleri*.

### 2.6. Spectral Pre-Processing and Data Analysis

Raman spectra were pre-processed using Opus software (Bruker optics GmBH, V 7.0, Massachusetts, United states of America) in the spectral ranges of 350–1800 cm^−1^ and 2800–3050 cm^−1^. The spectra were corrected using the elastic concave method (64° and 10 iterations), then smoothed twice using 25 smoothing points based on the Savitzky-Golay algorithm. Matlab software (version R2022a) and the SAISIR package were used for the subsequent steps [49]. The spectra were normalized by the standard normal variate (SNV) method and statistically analysed. All spectra were visualised using the Matlab ‘mesh’ function. To verify the spectral quality and identify repeatability and similarities, the spectra were correlated using the ‘corrcoef’ function. A kinetic study of the intensities of the main spectral bands was also performed. They were then examined by a principal component analysis (PCA), a descriptive statistical method producing a 3D PCA map whose axes consist of the principal component (PC) levels (score values) [50]. Based on the first PCA score, the spectra were analysed using a non-parametric one-way Kruskal–Wallis test, in which the groups were the days of culture.

## 3. Results

### 3.1. Microalgal Culture Monitoring at the Laboratory Scale

The 1793 spectra obtained over the 36 days of culture in the laboratory-scale 1-L photobioreactor revealed the kinetics of several molecules of interest (Figure 2A). The main bands are carbohydrates [δ (C-C-C)_479cm^−1^_]; phospholipids [C_4_N^+^, ν_s_ (O-C-C-N)_865cm^−1^_]; chlorophylls [δ (C-H_3_) _988cm^−1^_]; carotenoids [ν (C-C)_1157cm^−1^_ and ν (C=C)_1524cm^−1^_]; and lipids [σ (C-H_2_)_1444cm^−1^_, ν (C=C)_cis 1660cm^−1^_, ν (C=O)_1750cm^−1^_, ν (C-H_2_)_2850,2940cm^−1^_, ν (C-H_3_)_2885,2970cm^−1^_ and ν_as_ (=C-H)_3008cm^−1^_] [36,51,52,53,54] (Appendix A). The intensity of the carotenoid bands increased during the first days, followed by a gradual decrease, and then an increase again in the final days. The median spectra of days 0, 4, 8 and 32 show this kinetics of the bands (Figure 2B). According to the literature, nitrogen deficiency triggers several physiological changes, including an increase in intracellular lipids, a gradual cessation of biomass growth, and a decrease in pigment concentrations [55]. The results of the physical-chemical analyses confirm the physiological kinetics as described in the literature (Figure 2C). The figure shows how pigment concentrations increased from day 0 and 6, while nitrate concentration decreased. Appendix A presents the kinetics of culture density. The logarithmic values show an increase up until the 10th day, after which they stabilize.

### 3.2. Verification of Repeatability and Similarity of Raman Measurements

Correlations were calculated to evaluate the overall quality of the recorded spectra over the experimental phase of the laboratory-scale test (1793 spectra). Figure 3A shows a 2D map of the correlations of all the spectra collected, which can be interpreted using the table of correlation values obtained from the map. The physiological variations throughout the culture period are confirmed by a total average correlation of 0.81 (±0.11). However, these values varied according to the physiological state of the cells. During the first few days, there was a significant increase in pigment concentration, hence the cellular heterogeneity resulting in an average correlation of 0.91 (±0.07) [56]. From the sixth to the tenth day, the average correlation was 0.98 (±0.01). During these days, nitrate concentrations were low, growth stabilised and cellular heterogeneity decreased [57]. However, the correlation decreased after day 17 (0.83 ± 0.07). This observation is consistent with the triggered lipid production and various physiological changes [55]. During this period, the rate of dead cells also grew, increasing heterogeneity [58]. To further this analysis, the repeatability of the spectra was evaluated using the mean value of the correlation levels for each day as well as its standard deviation (Figure 3B). The standard deviation demonstrates the reproducibility of the spectra, thereby confirming their spectral quality. However, during the first days and from day 17 onwards, significantly higher spectral variability was observed, which aligns with the biological reality of microalgal cells exhibiting physiological heterogeneity during this phase. This heterogeneity in chemical composition can be detected through Raman spectroscopy [56,57,59]. Another strategy adopted was the comparison of spectra with a chosen reference day (the first day was selected as reference) (Figure 3C), which demonstrated that the microalgal cells deviated from this initial state and that heterogeneity increased over time. The 2720 spectra recorded during the pilot-scale phase were examined using the same correlation strategies to verify the spectral quality (Appendix A).

### 3.3. Validation of the Developed Approach at the Pilot Scale

The overview of the Raman spectra of *P. kessleri* grown at the pilot scale illustrates the changes in the chemical bands over time (Figure 4A). The bands corresponding to chlorophylls and carotenoids (e.g., 988 cm^−1^, 1157 cm^−1^ and 1524 cm^−1^) decreased dramatically through the culture (Figure 4B). On days 0 and 4, the chlorophyll and carotenoid bands were more intense than on the last days. Concerning lipid bands, the phospholipid band [C_4_N^+^, ν_s_ (O-C-C-N)_865cm^−1^_] showed weak intensity with slight variations between the days. Inversely, triglycerides [ν (C=O)_1750cm^−1^_] and unsaturated bonds [(C=H)_cis 1660cm^−1^_, ν (C-H_2_)_2850,2940cm^−1^_, ν (C-H_3_)_2885,2970cm^−1^_ and ν_as_ (=C-H)_3008cm^−1^_)] increased throughout the culture period.

The kinetics of the culture were also monitored by conventional analyses conducted in parallel with Raman spectrum analyses (Figure 4C). The concentrations of chlorophyll *a*, chlorophyll *b* and carotenoids stabilised from day 3. When the culture was fed with N-free nutrient medium on day 8, the pigment concentration decreased due to this nitrogen limitation, which initiated the growth of cellular reserve lipids [55]. In this phase, intracellular lipids increased from 27.77 to 48.85 mg·L^−1^ from day 7 to day 12. The lipid composition of the cells also changed. These lipid fractions of microalgal cells contain neutral lipids (NL), glycolipids (GL) and phospholipids (PL). On the one hand, phospholipids and glycolipids, which are conducive to cell growth and major constituents of the intracellular membrane, decreased over the course of the experiment. On the other hand, neutral lipids (such as triacylglycerols, TAGs), also known as reserve lipids, were present in higher concentrations in the later days. Analysing the fatty acid composition, saturated fatty acids (SFAs) were the most abundant form of fatty acids (FAs). The proportion of polyunsaturated fatty acids (PUFAs) and monounsaturated fatty acids (MUFAs) increased in relation to saturated fatty acids (SFAs). The proportion of the unsaturated form in the cell increased by 22.8%. Appendix A depicts the kinetic profile of culture density, demonstrating an increase in the absorbance values up to the 6th day. When an additional 50 L of BBM medium was introduced, the density decreased, subsequently increasing from the eighth day onwards. The results of the classical analyses on the Raman spectra were more clearly observed through the kinetic study of the intensity of the Raman bands. The ratio of the signal intensities of ν(C=C)_1660cm^−1^_ to δ(CH_2_)_1444cm^−1^_ is shown in Figure 5. This ratiometric method was used to estimate the content of SFA per fatty acid molecule without the influence of fluctuations and background noise [60]. The ratio of 1660 cm^−1^ to 1444 cm^−1^ suggests an increase in unsaturated bonds compared with saturated bonds over the whole culture time. Lipid bands at ν_as_ (C-H_2_) 2940 cm^−1^ and ν_as_ (=C-H) 3008 cm^−1^ confirm the increase of unsaturated fatty acids. Moreover, the band at 1750 cm^−1^ associated with TAGs also increased. Regarding carotenoids, the 1524 cm^−1^ band decreased its intensity over the culture production period. The ratio of 1157 cm^−1^ to 1524 cm^−1^ indicates that the variations of the ν(C-C) and ν(C=C) bonds remain proportional. These bands can be attributed to β-carotene, lutein or astaxanthin [61]. Although the total amount of carotenoids fluctuates, it is possible that the ratio between the carotenoid types did not change during the culture period.

### 3.4. Chemometric Analysis of the Spectral Database

A principal component analysis (PCA) was performed on the Raman spectra from the pilot-scale culture to generate a three-dimensional plane which we used to evaluate the distribution of spectra belonging to different physiological phases. The first principal component (PC1) accounted for 26.3% of the variance, while the PC2 and the PC3 accounted for 11.4% and 7.4%, respectively. The distribution in the three-dimensional space suggests two main sets of cells with differing physiology: a first set rich in pigments such as carotenoids and chlorophyll and a second set rich in storage lipids induced by low nitrate concentrations (Figure 6A).

The Kruskal–Wallis test based on the first PC (Figure 6B) produced 14 groups according to the days of culture. The *p*-value of =0 confirms that the day of culture has a significant effect. The Kruskal–Wallis plot shows that the median rank of the first 4 days (days 0 to 3) is distanced and consistent with an exponential growth phase and an increase in pigment concentrations in the culture. Then, day 3 and day 4 have a median rank without significant differences, but day 4 and day 5 have a significantly different median rank. As already seen in the previous analyses, the amount of nitrogen and the concentration of pigments decreased on this day. After this, the days were less differentiated until the 11th day, when the differentiation increased. These analyses were also applied to the spectra captured during the laboratory phase (Appendix A). The three-dimensional plan revealed a distinct distribution of groups, corresponding to the physiological state of this culture. The physiological phases were better defined, with PC1 accounting for 46.7% of the variability, and PC2 and PC4 accounting for 9.2% and 5.2%, respectively, cumulatively explaining the total variance of 75.4%. The Kruskal–Wallis test had a *p*-value = 1.40 × e^−284^, suggesting a significant difference among the production days.

## 4. Discussion

The present study offers a new methodology to allow the field use of Raman spectroscopy sensor in algal production process. The developed approach proposes the monitoring of microalgae cultures by a portable Raman spectrometer unlike the majority of publications which propose the use of confocal systems not suitable for field analysis [25,27,31,32]. The proposed approach is not limited to a comparison between the Raman spectra of the database and the spectra obtained in the field but offers an industrial solution to analyse algae in situ. This issue is economically viable due to the fully automated sampling station and the lightproof measurement chamber. In addition, the chamber has a standard dimension, can be operated with all commercial Raman probes and the used materials conform to the specifications outlined in the sanitary standards for industries [43]. Unlike other approaches which use specific systems and require specific setups for analysis, the automation of our sensor makes the system autonomous, reduces the possibility of human error and allows for the scheduling of culture analysis over a long timescale. The computer-controlled sampling station features many washing cycles with sterile water to reduce the risk of contamination and to prevent the formation of biofilms in the analysis area and/or the surface of the Raman probe [33,34]. The obtained Raman spectra can be used to distinguish the culture days and follow the evolution of spectral differences. Simultaneously, the information obtained from the experiments can be used to identify the optimal day for harvesting target molecules, as well as to quickly respond to the dysfunction of certain physicochemical parameters affecting cell physiology. Beyond this observation, the Raman sensor can monitor the kinetics of the molecules of interest, including saturated and unsaturated fatty acids and carotenoids, without any extraction or the use of chemicals. However, the proposed approach needs the implementation of new statistical models able to correlate between Raman fingerprints and the physicochemical analyses obtained from analytical methods. Currently, some of these models are in progress and will be addressed in a new scientific paper on data analysis. The main challenges will be the automation of the whole approach and the use of a single control interface with the chemometric models to simplify its use by an unqualified user. A reduction in the prices of spectrometers is also very important to help this technology become widely used.

## 5. Conclusions

This study proposes an analytical approach based on Raman spectroscopy to monitor the physiology of microalgae in situ. The proposed sensor consists of a fully automated sampling station and lightproof measurement chamber with standard dimensions which can be used with all commercial Raman probes. The computer-controlled station features many washing cycles with sterile water to prevent the formation of biofilms in the analysis area and the surface of the Raman probe. The effectiveness of the developed approach was successfully demonstrated in the laboratory and under pilot-scale conditions. The implementation of chemometric tools enabled the characterization of the biological dynamics on different culture days throughout the production period, demonstrating significant differences among these days and making it possible to identify physiological changes in the cells. The obtained Raman spectra can also be used to monitor the kinetics of molecules of interest in algal cells without any extraction or the use of chemicals. The proposed approach reduces the need for daily intervention and facilitates its integration in large-scale production. However, the main challenges will be the automation of the whole approach, including the chemometric models, to simplify its use by an unqualified user. A reduction in the prices of spectrometers is also very important to help this technology become widely used and it could be transferred to monitor many other bioprocess domains.

## Figures and Tables

**Figure 1 sensors-23-09746-f001:**
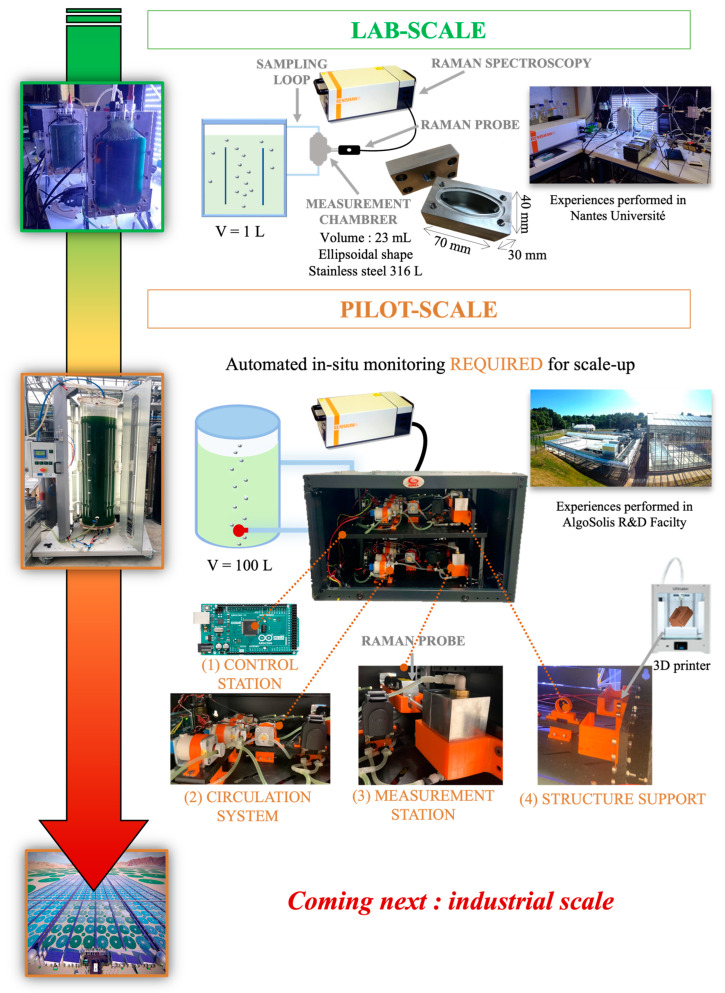
Development of a Raman measurement approach: from laboratory to pilot-scale applications.

**Figure 2 sensors-23-09746-f002:**
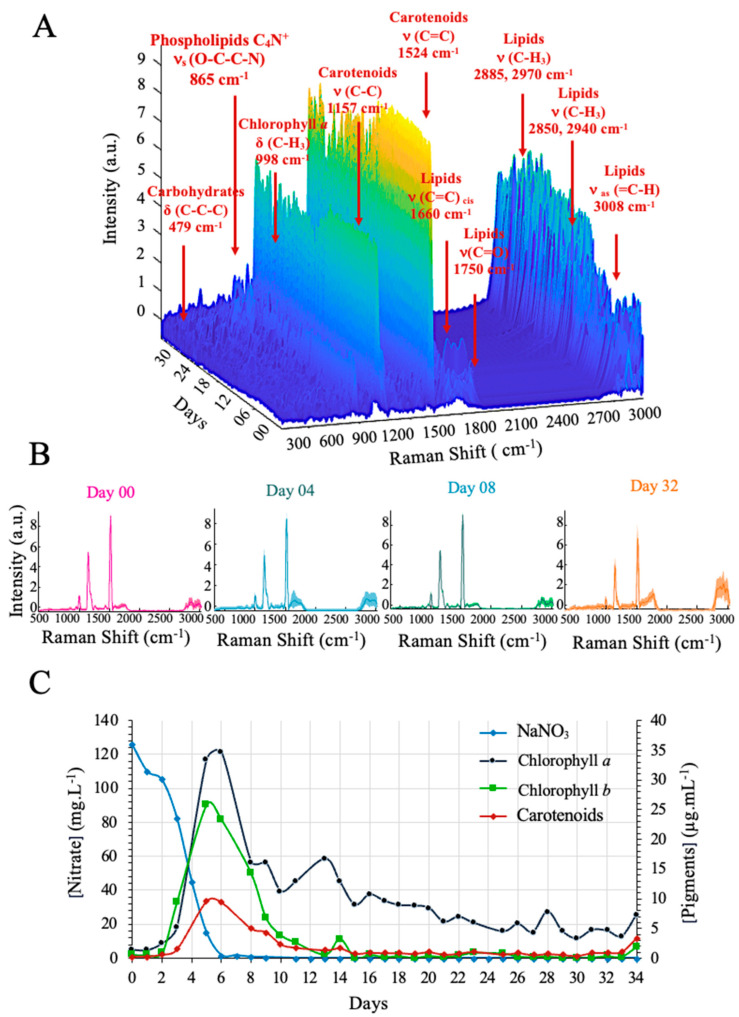
Monitoring of *Parachlorella kessleri* in Bold Basal Medium in a 1-L airlift photobioreactor. (**A**) Overview of 1793 spectra obtained over 36 days. (**B**) Median of 50 spectra recorded on day 0, day 4, day 8 and day 32. (**C**) Concentrations of chlorophyll *a*, chlorophyll *b* and carotenoids increase during cell growth until nitrogen starvation. Pigment concentration decreases when NaNO_3_ = 0 mg/L.

**Figure 3 sensors-23-09746-f003:**
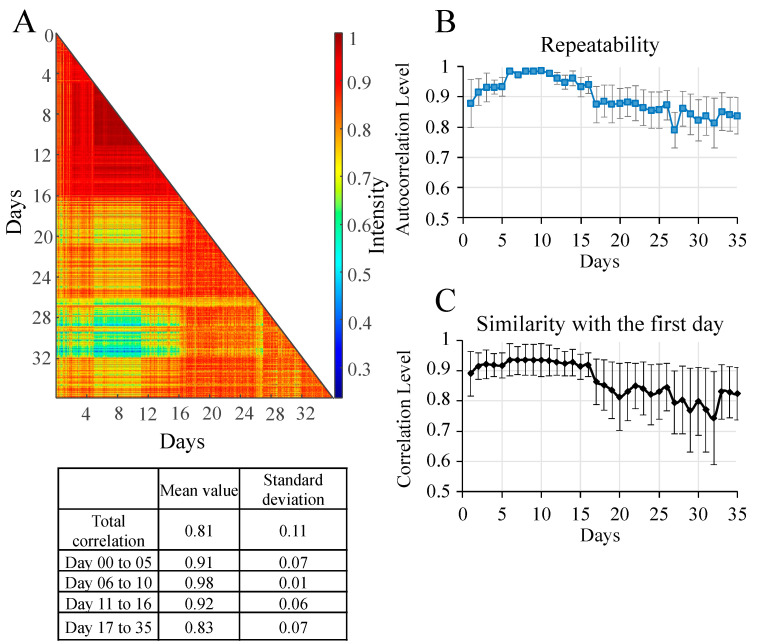
(**A**) 2D correlation map of all spectra with a correlation table covering the 36 days of culture in a 1-L airlift photobioreactor. The colour of each map point represents the level of correlation between two spectra, from red (highest correlation) to blue (lowest correlation). (**B**) Repeatability of spectra measured by the autocorrelation level between 50 spectra recorded in the same time window. (**C**) Similarity of the spectra to those of the first day assessed by calculating the correlation between them.

**Figure 4 sensors-23-09746-f004:**
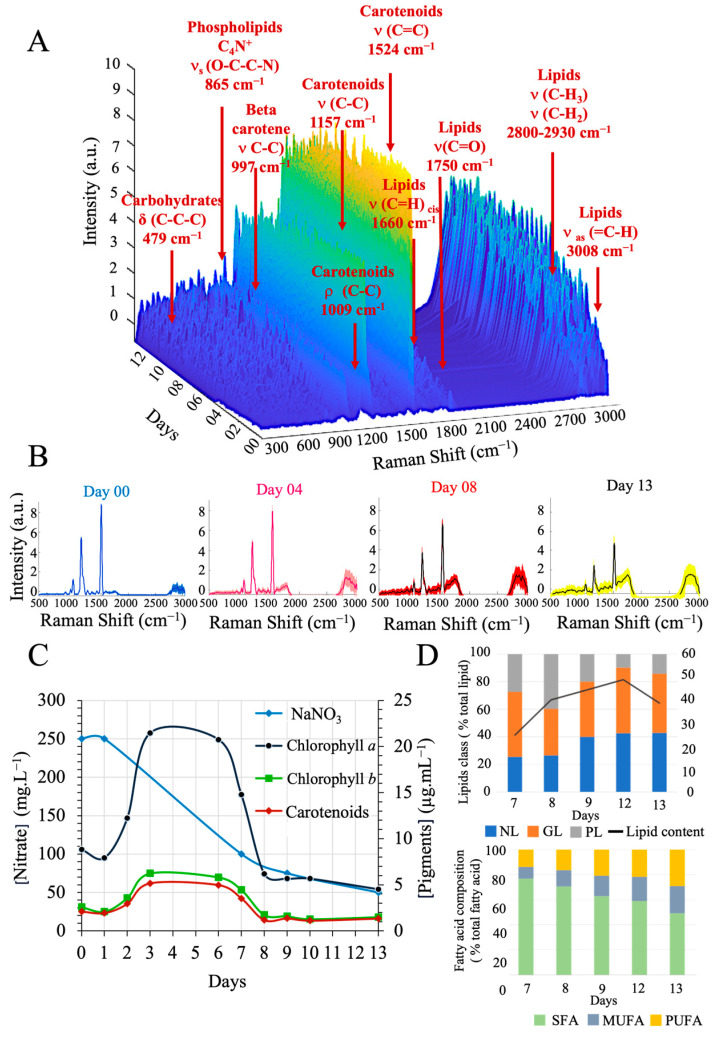
Monitoring of *Parachlorella kessleri* in Bold Basal Medium in a 100-L tubular airlift photobioreactor from day 0 to day 13. (**A**) Overview of 2720 spectra obtained at 13 days. (**B**) Median of 50 spectra recorded on day 0, day 4, day 8 and day 13 (**C**) Concentrations of chlorophyll *a*, chlorophyll *b* and carotenoids increase during cell growth until nitrogen limitation. After significant nitrogen limitation, the concentration of pigments decreases. (**D**) Lipid analyses of samples collected between day 7 and day 13. Abbreviations indicate: NL, neutral lipids; GL, glycolipids; PL, phospholipids.

**Figure 5 sensors-23-09746-f005:**
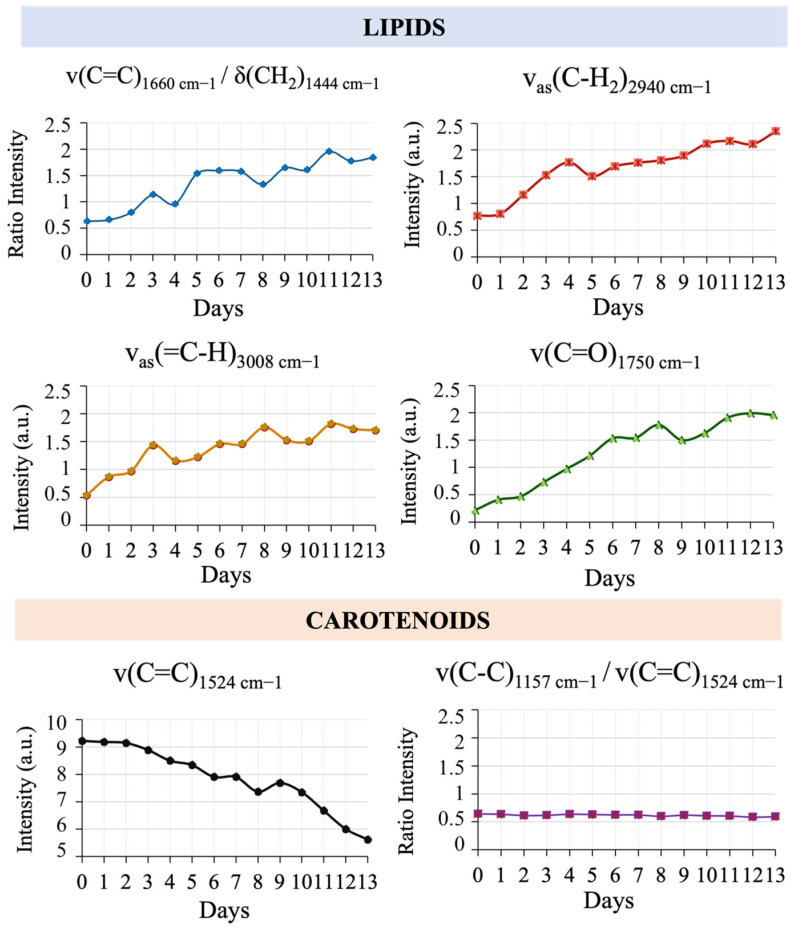
Raman band intensity over 14 days at pilot scale in a 100-L photobioreactor: v(C=C)_1660cm^−1^_/δ(CH_2_)_1444cm^−1^_, v_as_(C−H_2_)_2940cm^−1^_, v_as_ (=C−H)_3008cm^−1^_, v(C=O)_1750cm^−1^_, v(C=C)_1524cm^−1^_ and v(C−C)_1157cm^−1^_/v(C=C) _1524cm^−1^_.

**Figure 6 sensors-23-09746-f006:**
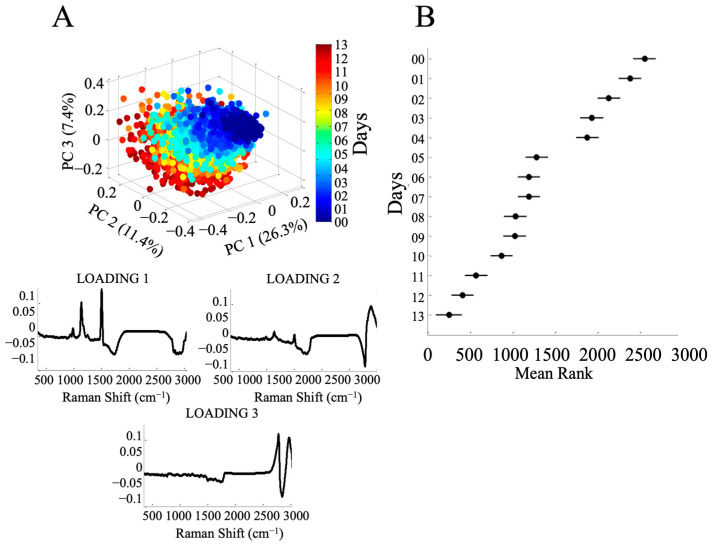
(**A**) Three-dimensional representation of the principal component analysis (PCA) of spectra from a 14-day culture in a 100-L tubular airlift bioreactor (PC1 26.3%; PC2 11.4%; PC4 7.4%) and their three respective loadings. (**B**) Kruskal–Wallis one-way ANOVA test, based on PCA loading 1, representing the variance of the spectra over the 13 days.

## Data Availability

Not applicable.

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
