# Peer review of "Development and Application of an Automated Raman Sensor for Bioprocess Monitoring: From the Laboratory to an Algae Production Platform"

_sensors, 2023, doi:10.3390/s23249746_

Round 1

Reviewer 1 Report

Comments and Suggestions for Authors

1. The format of the second-level headings and third-level headings at lines 95, 134, 152, 184, etc. is incorrect.

2. It is necessary to modify the formula and the reference of the formula. The format of equation (1), equation (2), equation (3) is wrong, and the reference format of equation (4) is wrong.

3. How to ensure that your experiments are reproducible? When there are differences in the samples obtained each time, the current analysis will become unsuitable.

4. Was the Raman spectroscopy instrument used in the test provided by the instrument supplier? What challenges will it face in the later pilot trials and industrialization?

5. Lack of discussion and conclusion.

6. The grammar of the paper needs to be modified to some extent.

Comments on the Quality of English Language

The grammar needs modest revision.

Reviewer 2 Report

Comments and Suggestions for Authors

I would like to express my gratitude to the authors for their valuable work. The manuscript deals with the introduction of an automated Raman spectroscopy-based sensor for the monitoring of microalgal production. It addresses the response under diverse culture conditions, offering insights into the physiological state of microalgal cells, as well as monitoring intracellular molecules relevant to production settings. The manuscript displays a commendable structure, explanations, and a suitable number of figures both in the manuscript and its accompanying supplementary materials. To enhance the quality of the study, I propose the following points for the authors' consideration:

1. The authors should make a concerted effort to explicitly highlight the novelty of their work, underlining distinctions from existing literature.

2. In section 2.2, it is essential to provide a comprehensive explanation of the measurement techniques and a clear justification for the factors employed in the equations. Additionally, I recommend updating the references related to determination methods. Failing that, the authors should explore the validation of these methods.

3. Regarding the pilot-scale study, it would be advantageous for the authors to elucidate how they ensured the isolation of cultures from a single algal strain. Alternatively, they should investigate the possibility of a mixed culture, which could potentially exhibit distinct metabolic activities compared to Parachlorella kessleri.

4. I would like to emphasize the existence of pertinent literature concerning the control of algae and microalgae cultures. Specifically, the references cited include https://doi.org/10.1016/j.chemolab.2011.09.007, https://doi.org/10.1021/ac050281z, https://doi.org.10.1039/C3AN01158E, and https://doi.org/10.1080/07388551.2017.1398132. I encourage the authors to find ways to differentiate their work from these studies and accentuate the strengths and benefits of their approach.

In light of these considerations, I recommend rejecting the manuscript in its current form for publication. However, I remain open to re-evaluating it once the suggested changes have been implemented.

Reviewer 3 Report

Comments and Suggestions for Authors

In this manuscript, an online monitoring sensor for microalgae production based on automatic matching Raman spectroscopy is proposed. An in-situ system with a sampling station consists of an opaque optical chamber connected to a Raman probe. The microalgae culture is transported to this chamber through a pipeline connected to a pump and computer-controlled and programmable valves. Some revisions are needed before publication in Sensors.

Some detail information need to be clarified :

1. In the introduction section, the introduction should be more summarizing and the development process should be more brief.

2.During the experimental data collection stage, it seems that the impact of different distribution positions of algae was not considered, and additional information is needed.

3.When determining the concentration of pigments, it should be explained why the absorbance at 480 nm (A480), 652 nm (A652), and 665 nm (A665) was selected.

4.Figures 2. C and 4. C both indicate that the concentrations of chlorophyll a, chlorophyll b, and carotenoids increase during cell growth. It should be explained why the platform period appears in Figure 4. C.

Round 2

Reviewer 1 Report

Comments and Suggestions for Authors

The author made extensive revisions to the questions raised and supplemented them with experiments. The article has good language expression, clear research ideas, and good practical value. It is recommended to revise the article before accepting it.

Comments on the Quality of English Language

The language expression is good, it is recommended to check and modify the language again to ensure there are no issues.

Reviewer 2 Report

Comments and Suggestions for Authors

I appreciate the authors' courtesy in addressing the suggestions I made in my previous review. The manuscript has been significantly enriched, and the clarifications provided by the authors are satisfactory. I recommend accepting the current manuscript for publication.